

# Artificial intelligence-assisted air quality monitoring for smart city management

En Xin Neo[1], Khairunnisa Hasikin[1,2], Khin Wee Lai[1], Mohd Istajib Mokhtar[3], Muhammad Mokhzaini Azizan[4], Hanee Farzana Hizaddin[5], Sarah Abdul Razak[6] and Yanto[7]

[1] Department of Biomedical Engineering, Faculty of Engineering, Universiti Malaya, Kuala Lumpur, Malaysia
[2] Center of Intelligent Systems for Emerging Technology (CISET), Faculty of Engineering, Kuala Lumpur, Malaysia
[3] Department of Science and Technology Studies, Faculty of Sciences, Universiti Malaya, Kuala Lumpur, Malaysia
[4] Department of Electrical and Electronic Engineering, Faculty of Engineering and Built Environment, Universiti Sains Islam Malaysia, Nilai, Negeri Sembilan, Malaysia
[5] Department of Chemical Engineering, Faculty of Engineering, Universiti Malaya, Kuala Lumpur, Malaysia
[6] Institute of Biological Science, Faculty of Science, Univerisiti Malaya, Kuala Lumpur, Malaysia
[7] Civil Engineering Department, Jenderal Soedirman University, Purwokerto, Indonesia

Corresponding author
Khairunnisa Hasikin,
khairunnisa@um.edu.my

## ABSTRACT

**Background**. The environment has been significantly impacted by rapid urbanization, leading to a need for changes in climate change and pollution indicators. The 4IR offers a potential solution to efficiently manage these impacts. Smart city ecosystems can provide well-designed, sustainable, and safe cities that enable holistic climate change and global warming solutions through various community-centred initiatives. These include smart planning techniques, smart environment monitoring, and smart governance. An air quality intelligence platform, which operates as a complete measurement site for monitoring and governing air quality, has shown promising results in providing actionable insights. This article aims to highlight the potential of machine learning models in predicting air quality, providing data-driven strategic and sustainable solutions for smart cities.

**Methods**. This study proposed an end-to-end air quality predictive model for smart city applications, utilizing four machine learning techniques and two deep learning techniques. These include Ada Boost, SVR, RF, KNN, MLP regressor and LSTM. The study was conducted in four different urban cities in Selangor, Malaysia, including Petaling Jaya, Banting, Klang, and Shah Alam. The model considered the air quality data of various pollution markers such as $PM_{2.5}$, $PM_{10}$, $O_3$, and CO. Additionally, meteorological data including wind speed and wind direction were also considered, and their interactions with the pollutant markers were quantified. The study aimed to determine the correlation variance of the dependent variable in predicting air pollution and proposed a feature optimization process to reduce dimensionality and remove irrelevant features to enhance the prediction of $PM_{2.5}$, improving the existing LSTM model. The study estimates the concentration of pollutants in the air based on training and highlights the contribution of feature optimization in air quality predictions through feature dimension reductions.

**Results**. In this section, the results of predicting the concentration of pollutants ($PM_{2.5}$, $PM_{10}$, $O_3$, and CO) in the air are presented in $R^2$ and RMSE. In predicting the $PM_{10}$ and $PM_{2.5}$ concentration, LSTM performed the best overall high $R^2$ values in the four study areas with the $R^2$ values of 0.998, 0.995, 0.918, and 0.993 in Banting,

Petaling, Klang and Shah Alam stations, respectively. The study indicated that among the studied pollution markers, $PM_{2.5}$, $PM_{10}$, $NO_2$, wind speed and humidity are the most important elements to monitor. By reducing the number of features used in the model the proposed feature optimization process can make the model more interpretable and provide insights into the most critical factor affecting air quality. Findings from this study can aid policymakers in understanding the underlying causes of air pollution and develop more effective smart strategies for reducing pollution levels.

## INTRODUCTION

Our environment has been greatly impacted by rapid urbanization in terms of toxins and signs of climate change. There are serious threats posed by climate change and global warming has affected both developed and developing countries. This alarming situation has arisen due to the rise of economic activities, prompted by urbanization needs and life quality enhancements. With the urban space expansion, increasing use of industrial technology, and expansion of transportation sectors, urban air pollution has become one of the byproducts of rapid urbanization. From a health perspective, urban air pollution has led to severe health hazards (*Saini, Dutta & Marques, 2020*), not to mention the ill effects of climate change, its influence on the atmospheric environment, and disruptive changes in the ecosystem (*Han et al., 2019*). According to a recent study by the Global Burden of Disease project, poor air quality causes the early mortality of 5.5 million people globally per annum (*GBD 2013 Risk Factors Collaborators, 2015*). The study indicated that the effect is adversely impacted by the quality of air surrounding us, and therefore having clean air is very important in extending the life span. The low quality of air has led to several health complications such as respiratory diseases, cardiorespiratory diseases, various types of cancers, and pregnancy and birth complications (*Neo et al., 2022*).

Malaysia as one of the developing countries is not excluded from facing the grave threat of climate change and global warming. In 2022, Malaysia (4.1205° N, 101.9758° E) total population is 33,871,431 with the state of Selangor has the largest population of 7.9 million people compared to the others. This made each of the districts in Selangor an urban area as shown by its population distribution in each district in Fig. 1. From the perspective of gross domestic product (GDP), Selangor contributed the growth of approximately 5% in 2021 as compared to the previous year, with a value-added of RM 343.5 billion ($76.92 billion) in 2021, as compared to 2020 which is only RM 327.1 billion ($73.25 billion) (*Department of Statistic, 2022*; *Invest Selangor, 2022*), making up almost a quarter of contribution (24.32%) to national GDP up to the year 2020. Numerous studies demonstrated that air pollution was most prevalent in large cities with high seasonal heating demands, heavily industrialization, high vehicular traffic volumes or a

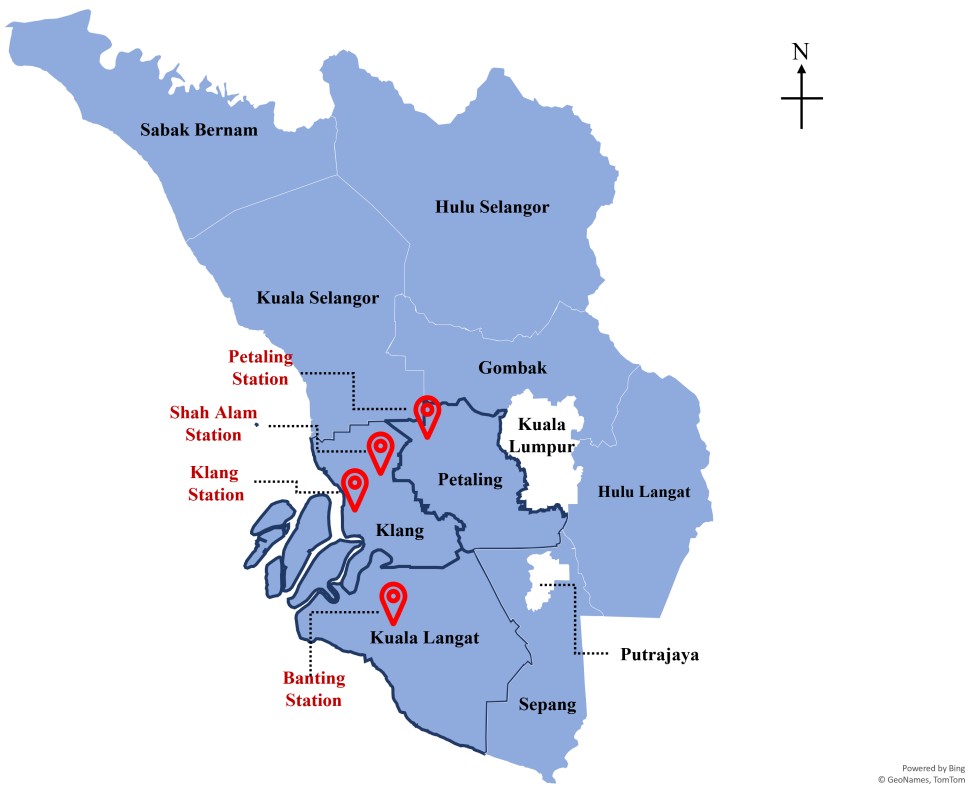

**Figure 1** Population distribution and air pollution monitoring stations location in Selangor. Map credit: ©GeoNames TomTom.

combination of all three (*Shen, Valagolam & McCalla, 2020*; *Yin et al., 2021*; *Zhang et al., 2021*).

According to the United Nations, the global population is in a growing trend with an approximately 2.34 billion population reside in Eastern and South-eastern Asia (*United Nations Department of E, and Social Affairs PD, 2022*). Meanwhile in Malaysia, in the third quarter of 2022, its population reached 32.9 million continuing a positive growth pattern (*Department of Statistic, 2022*). As a result of rapid expansion and a large population, Malaysia, especially Selangor has spontaneously encountered the highest pollution problems, particularly air pollution.

Air pollution contributed to severe health impacts including cardiorespiratory diseases, prenatal complications and premature mortality, cancers and increase hospitalization rate (*Achebak et al., 2021*; *Al Noaimi et al., 2021*; *Peng et al., 2020*; *Reid et al., 2019*; *Tusnio et al., 2020*; *Usmani et al., 2021*; *Zani et al., 2020*). A collaboration between government agencies is crucial not only in monitoring the pollution levels but also in preventing severe health impact. However, predicting air pollution is challenging due to the influence of multiple factors, such as different spatial and temporal distribution and specific factors that can cause a sudden change in air quality index, which in turn affects the monitoring of air quality.

Therefore, there is an urgent need to develop a predictive system with the aid of artificial intelligence (AI) for smart air quality monitoring. The AI-assisted system can play an important role for the government or authorities to anticipate air pollution, maintaining its quality and assessing its impact towards achieving low carbon city. In order to accomplish this goal, determining the interaction patterns between pollutants was an immediate necessity. Current practice in monitoring air quality is done by quantifying pollutant concentrations through the installed sensor networks in the monitoring stations. The concentrations of these pollutants are then remotely monitored to ensure that they are below the World Health Organization (WHO) and United State Environmental Protection Agency (USEPA) threshold levels. Thus, AI can play a role in expanding the existing air pollution monitoring network, for instance by interpreting the sensor devices' measurement signals. If used in conjunction with measurements from exiting monitoring stations, such devices may be utilized to fill monitoring gaps.

Currently, interpretation and forecasting of air pollution requires complex numerical models that simulate weather and air pollution chemistry. The combination of low-cost air pollution sensors with artificial intelligence and hybrid models may offer the potential for much more detailed air pollution maps and, consequently, better-targeted mitigation measures than are currently available. In combination with physiological sensors and medical information systems, AI based pollution monitoring may eventually enable direct measurements of inhaled pollutant doses, enabling vulnerable individuals to more effectively plan outdoor activities and avoid hazardous environments.

## LITERATURE REVIEWS

Unquestionably, the role of technological progression in reducing air pollution and enhancing environmental quality is interdependent. *Wang et al. (2021)* discussed the causes of air pollution in government-led economies based on the behavior of the prefectural government's annual economic growth projection. The findings indicate that the local government's ambitions for economic growth will effectively support economic growth, despite the risk of a significant increase in air pollution. Therefore, *Wang et al. (2021)* proposed that by enhancing green development-related indicators, the central government should incorporate environmental development, green development capabilities, and other sustainable development criteria into the official assessment system as one solution to the aforementioned problems. In this manner, local government should collaborate to manage air pollution and improve environmental quality, while providing a long-term incentive mechanism for green technology innovation in business (*Wang et al., 2021*).

It is a monumental challenge for us to overcome the issues and complications from urbanization in the era of the second digital and fourth industrial revolution (IR 4.0). Through the use of tools and leveraging of technologies, the demand for automated devices and computing resources has increased (*Idris et al., 2014*). Consequently, raising the trend of the concept of smart cities towards the emergence of AI and IR4.0 nowadays. Smart city is gaining popularity as a result of the industry transformation utilizing integration and intelligent engineering (*Muhuri, Shukla & Abraham, 2019*). The smart city

is viewed as the location where digital technology and data are widely utilized to produce efficiency for sustainability, quality of life, and economic growth (*Mora, Deakin & Reid, 2019*). AI is undoubtedly has become the topic of discussion on the transition of cities into smart cities in many urban policy circles, especially among urban policymakers and planners who look for a technocentric answer to grave urbanization issues (*Kassens-Noor & Hintze, 2020*). AI is a disruptive technology with a wide range of applications and vast future potential in every industrial sector and aspect of daily life, including engineering, finance, gaming, health, agriculture, and transportation (*Cugurullo, 2020*). Today, global smart city initiatives are primarily driven by AI (*Singh et al., 2020*). This appeal is attributable to the growing acceptance of technocentric solutions as viable solutions to the numerous and complex problems associated with urbanization including quality of life, climate change safety and security, mobility, and accessibility (*Yigitcanlar et al., 2020*). It is anticipated that big data, AI-powered smart urban technologies, and platforms will improve the efficiency of urban services and infrastructure as well as address or substantially reduce the challenges (*Corchado et al., 2021*; *Yu & Zhang, 2019*). AI is crucial because it is one of the foundational technologies in the age of data and digitalization, especially in IR4.0. AI is frequently employed in the field and research of medical and environmental sustainability research (*Jamaludin et al., 2022*; *Mammoottil et al., 2022*; *Teoh et al., 2022*; *Woan Ching et al., 2022*; *Wong et al., 2022a*; *Wong et al., 2022b*; *Yeoh et al., 2021*). As smart cities emerge, this article proposed the viability of AI in providing technological solutions to urban environmental problems, particularly the forecasting and control of urban air quality. We summarized several studies that have been implementing AI in air pollution monitoring in smart cities as tabulated in Table 1.

Therefore, based on the research gaps highlighted in Table 1. Our research proposed a comprehensive framework for predicting and managing urban air quality using AI-assisted technology of machine learning. Findings from this study will aid policymakers in better apprehending the air quality and thus provide smart management and enforcement. Overall, the main contributions of this study are as follows:

- The majority of past research has been on forecasting specific air pollution concentrations, such as $PM_{2.5}$, or $O_3$, using numerous variables, such as other greenhouse gas emissions (CO, NOx, *etc.*) as well as meteorological and precipitation data. However, the interrelation between various pollutant markers remained complex, and the effect of population growth on the overall assessment of $PM_{2.5}$ concentration has not been adequately explored. Determining the most critical parameters necessitated an assessment of the pollution markers and quantification of their linkages. By anticipating the most important parameters in projecting $PM_{2.5}$ concentration, policymakers and the government would be aided in their monitoring and enforcement efforts.
- Secondly, the related studies in Table 1 have utilized many input parameters or features in producing a prediction or forecasting model with higher accuracy. Therefore, our study presented feature optimization techniques motivated by the ability of removing unimportant features in the training stage using sensitivity analysis. We believe that the techniques may enhance the development of the prediction models

**Table 1  Summary of literature in implementation of IR4.0 concept and AI application in air pollution predictions.**

| Authors | Techniques | Input and Output Parameters | Research Emphasis | Result | Research Gap |
|---|---|---|---|---|---|
| *Celis e al. (2022)* | SVM, LSTM, Bidirectional LSTM | **Input and Output:** PM2.5 | $PM_{2.5}$ behaviour using machine learning model and early alert system based on risk probability forecast | Bidirectional LSTM demonstrated the most precise prediction and performances. | More ANN structure testing is required for steeper changes, to improve performance. Improvement in robustness of risk assessment calculation for $PM_{2.5}$ impacts. |
| *Gao & Li (2021)* | GLSTM | **Input:** $PM_{2.5}$, $PM_{10}$, CO, $O_3$, $NO_2$, $SO_2$, Temperature, pressure, humidity, wind speed and precipitation **Output:** $PM_{2.5}$ | $PM_{2.5}$ concentration prediction using graph neural network and LSTM (GLSTM) model. | GLSTM able to realize the synchronous calculation and reduce workload in all stations. And able to predict overall $PM_{2.5}$ change over time. | Uncertainties of correlations between meteorological and air quality stations. Heterogeneous graph neural network should be considered in $PM_{2.5}$ prediction. |
| *Kow et al. (2022)* | CNN, LSTM, hybrid of multiple CNN and BPNN (MCNN-BP), CNN-LSTM-BP, weather research and forecasting-chemistry (WRF-Chem) | **Input:** $PM_{2.5}$, $PM_{10}$, CO, $O_3$, $NO_2$, $SO_2$, humidity, temperature **Output:** $PM_{2.5}$ | Prediction of $PM_{2.5}$ and the occurrence of air pollution and time-lag phenomena using MCNN-BP model (multi convolutional and backpropagation neural networks) | MCNN-BP model demonstrated shorter computational time and lower computational load while offers satisfactory $PM_{2.5}$ forecast. | Research methodology could be improved to increase learning efficiency and model accuracy. Real-time satellite resolution should be improved for predictive accuracy and interpretability. |
| *Lu et al. (2021)* | LSTM-RNN, Random forest, Lasso, WRF-CMAQ model | **Input:** $PM_{2.5}$, $PM_{10}$, CO, $O_3$, $NO_2$, $SO_2$, pressure, temperature, dew point temperature, humidity, wind direction wind speed and precipitation. **Output:** $O_3$ | Ozone forecasting using machine learning method, lasso and random forest involved for feature selection. | LSTM-RNN demonstrated relatively satisfactory prediction. | The dataset scope could be expand (select more than 1 year data) to enhance the performance. Possibility of other pollutants prediction using this model. |
| *Ma et al. (2020)* | WRF-Chem, XGBoost | **Input:** $PM_{2.5}$, $PM_{10}$, CO, $O_3$, $NO_2$, $SO_2$, Temperature, pressure, humidity, wind direction, wind seed, precipitations. **Output:** $PM_{.2.5}$ | Development of $PM_{2.5}$ prediction model using XGBoost algorithm and Lasso linear regression based on WRF-Chem outputs and air pollutant and meteorological observations. | XGBoost improves the $PM_{2.5}$ prediction accuracy of WRF-Chem model. XGBoost can predict winter heavy pollution with high accuracy. | $PM_{2.5}$ is forecasted daily, and monthly. Hourly prediction should perform. Larger study area and scope should be considered to increase reliability and accuracy. |

**Table 1** (*continued*)

| Authors | Techniques | Input and Output Parameters | Research Emphasis | Result | Research Gap |
|---|---|---|---|---|---|
| *Wu, Liu & Duan (2020)* | Eensembel model: linear programming boosting (LPBoost), combined with several outlier robust extreme learning machine (ORELMS) | **Input:** $PM_{2.5}$, $PM_{10}$, CO, $O_3$, $NO_2$, $SO_2$, AQI **Output:** $PM_{2.5}$ | $PM_{2.5}$ concentration forecasting model for early warning information system for air pollution exposure and public health monitoring. | The proposed model is proved to be effective, improve the forecasting performance and capacity. The $PM_{2.5}$ pollution prediction can be performed in real time for early warning system. | The possibility of model utilization in rural area is lacking. Data accuracy and reliability in real time prediction. Input parameters without meteorological factors. |
| *Schürholz, Kubler & Zaslavsky (2020)* | LSTM | **Input:** $PM_{2.5}$, $PM_{10}$, CO, $O_3$, $NO_2$, $SO_2$, temperature, humidity, wind speed, wind direction, traffic, fire incident, geo-location, age, user-id, pollutant sensitivity **Output:** AQ | Prediction of air quality and health conditions effect through context-aware integrated model using LSTM, making situation-specific event aware system. | High precision of 90–96% in forecasting air quality and model is highly adapted to user's health condition. | Possible and potential pollution sources are not specifically identified, such as natural phenomena. Uncertainties of real-life and real-time data sources due to malfunction measuring stations and disturbance. |
| *Honarvar & Sami (2019)* | Neural network and regression | **Input:** Traffic, location. O3, CO, SO2, NO2, PMs, temperature, dew point, humidity, sea level, visibility, wind, precipitation **Output:** PM10 | Prediction of PM based on transfer learning concepts, without using air pollution sensors. | The proposed predictive model is proven on its efficiency on air quality prediction intral time. | Possibility of utilizing social datasets such as social event and social media data in improving air quality estimation accuracy. |
| *Mihăiţă et al. (2019)* | decision trees, and neural networks on mobile air quality | **Input:** NO2, humidity and temperature, precipitation, and wind **Output:** NO2 | Air quality evaluation and prediction using sensors (fixed and mobile) and machine learning methods. | Both decision ree and neural network predict air quality accurately | Data accuracy issue. Data verification under different traffic and weather conditions is required in the evaluation for air quality monitoring accuracy. |
| *Saheer et al. (2022)* | ARIMA, linear regression, support vector regression, LSTM | **Input:** $PM_{2.5}$, $PM_{10}$, CO, $NO_x$, $NO_2$, $SO_2$, temperature, dew point temperature, wind speed, wind direction, pressure, rain, maximum wind speed, sunshine hours, vegetation. Satellite images **Output:** NO2 and PM10 | Data-driven $NO_2$ and $PM_{2.5}$ prediction integrated with weather conditions and vegetation information. | ARIMA shows slightly better performance but limitted to time series trend. LSTM do not show significant performance but as the potential to be tuned with bigger data size and parameter optimization. | Only 2 pollutants are involved as prediction target. Other types of pollutant should be included to understand various features affecting the pollutant concentrations. Tree species or vegetation information should be incorporated for emissions estimation. |

Neo et al. (2023), *PeerJ Comput. Sci.*, DOI 10.7717/peerj-cs.1306

**Table 1** (*continued*)

| Authors | Techniques | Input and Output Parameters | Research Emphasis | Result | Research Gap |
|---------|-----------|----------------------------|-------------------|--------|--------------|
| *Zhou et al. (2018)* | Linear regression Logistic Regression, support vector regressor, non-linear autoregressive neural network | **Input:** $PM_{2.5}$, $PM_{10}$, CO, $O_3$, $NO_2$, $SO_2$, temperature, wind speed, wind direction, rainfall, pressure, humidity, solar radiation **Output:** AQI | Air quality index prediction using environmental monitoring and meteorological data, by implementing enhanced non-linear autoregressive neural network | NARX achieve good performance in AQI and pollution prediction without $PM_{10}$. LR is better to predict AQI and pollution prediction with $PM_{10}$ | Neural network-based model is not optimum in $PM_{10}$ prediction. Correlations between sensing site location, and across different cities and AQI patterns and analysis of influences of environmental factors such as traffic and green covers yet to discovered. |

by reducing the amount of data required from the air quality monitoring station for prediction analysis. For instance, air quality prediction models may include various features such as temperature, humidity, wind speed, traffic volume, and many others. By performing feature optimization, the model can identify which features have the strongest impact on air quality, and which features can be excluded without significantly affecting prediction accuracy.

- In addition, there are growing concerns from previous research that emphasized on the impact of air pollution with regards to the population growth and urbanization activities. This is especially concerning developing countries such as Malaysia to accurately quantify the spatial and temporal variations of urban pollution emissions. Therefore, through this study, the correlation between pollutant markers is determined and thus will aid policy makers and government agencies in identifying the root cause of the pollution spreads.

## MATERIALS & METHODS

To further depict the situation of urbanization's impact on overall air quality, we selected Selangor, the Malaysian state with the highest GDP, as a case study. Selangor is located on the west coast of Peninsular Malaysia, encircling the capital Kuala Lumpur, with a geographical coordinate of 3.078° N, 101.5183° E. Selangor is a well-known investment heaven with a well-established infrastructure for major industry clusters, strong state government support and an innovative commercial ecosystem. Selangor is expected to have a GDP of approximately RM 343.5 billion ($76.92 billion) in 2021 making it the most economically significant state in Malaysia. As a result, Selangor has shown rapid growth in population and thus causing Selangor's air quality to be significantly worse than the other states. Due to the expansion of the economy, there is a growing need to monitor and predict the air quality in Selangor in order to prevent air pollution caused by its byproducts.

In this study, the data used in the prediction of air quality are collected from the air quality monitoring stations as depicted in Fig. 1. The air pollution datasets used to develop the air quality prediction model were obtained from the Department of Environment (DoE), Malaysia. The datasets were collected from the four air quality monitoring stations namely Petaling Jaya, Shah Alam, Klang and Banting (Fig. 1). The data is collected on an hourly basis from 2010 to 2016. On this study, the air quality parameters that are included are $PM_{10}$, $PM_{2.5}$, $O_3$, CO, $SO_2$ and $NO_2$, and the meteorological parameters such as wind direction, wind speed, humidity, and temperature. The description and characteristics of the datasets will be explained in the Dataset subsection. The preparation of datasets for predictions will be described in the Method subsection.

### Dataset

The data utilized in this study were collected from the Department of Environment, DoE Malaysia. The gathered data are retrospective hourly data acquired between 2010 and 2016. From this duration, four monitoring stations collected approximately 24,547,200 data per hour for 10 parameters which include ozone ($O_3$), particulate matter ($PM_{10}$

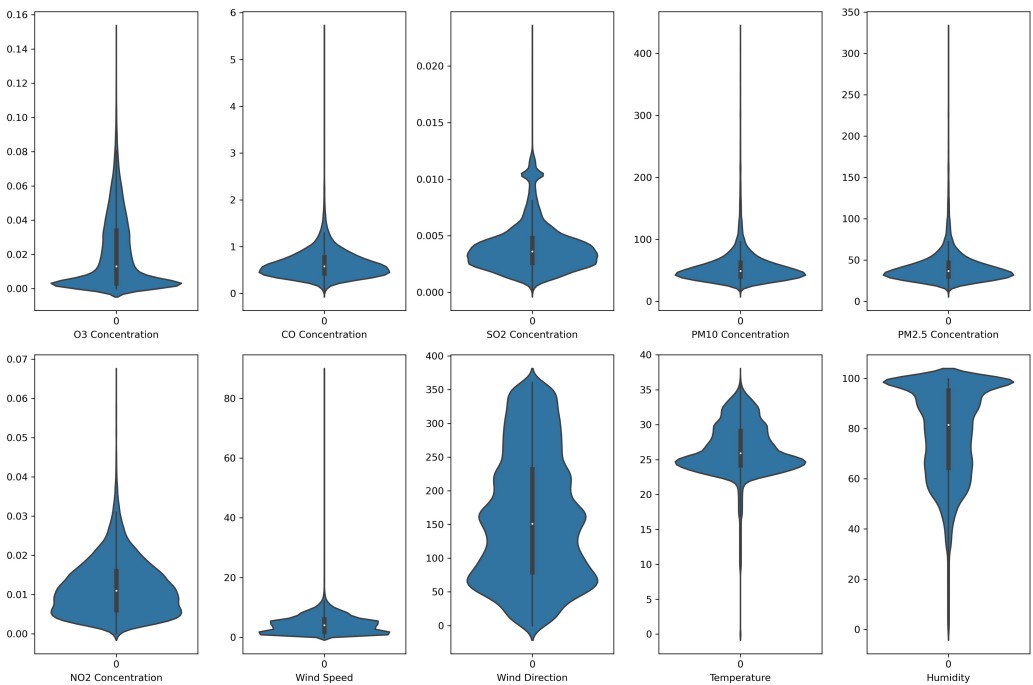

**Figure 2** Violin plot of the pollutants and meteorological data distribution in Banting air monitoring station.

and PM$_{2.5}$), nitrogen dioxides (NO$_2$), sulfur dioxides (SO$_2$) and carbon monoxide (CO), temperature, humidity, wind speed and wind direction.

## Air pollutant markers and meteorological observations

In this study, each monitoring station's data is unique. Each pollutant marker exhibited a distinct spatiotemporal distribution in various locations. In addition, due to the diverse populations and socio-economic activities in the regions, the pollutant markers and their concentration readings also differ accordingly. In this case, the violin plot is excellent for illustrating the distribution and characteristics of the dataset. In this study, there are six pollution parameters and four meteorological parameters in the study which include ozone (O$_3$), particulate matter (PM$_{10}$ and PM$_{2.5}$), nitrogen dioxides (NO$_2$), sulfur dioxides (SO$_2$), and carbon monoxide (CO), temperature, humidity, wind speed and wind direction were collected hourly from Department of Environment (DoE), Malaysia, at each station. Figures 2 to 5 depict the violin plot of the pollutants data distribution in the air monitoring stations in Selangor. Meanwhile, Table 2 illustrated the statistical analysis of the parameters in the 4 monitoring stations in Selangor.

## Parametric correlation

To identify the most significant parameters affecting the predictive model development, the Pearson correlation coefficient technique as in the equation below has been adopted to quantify the linear association between the input parameters in this study.

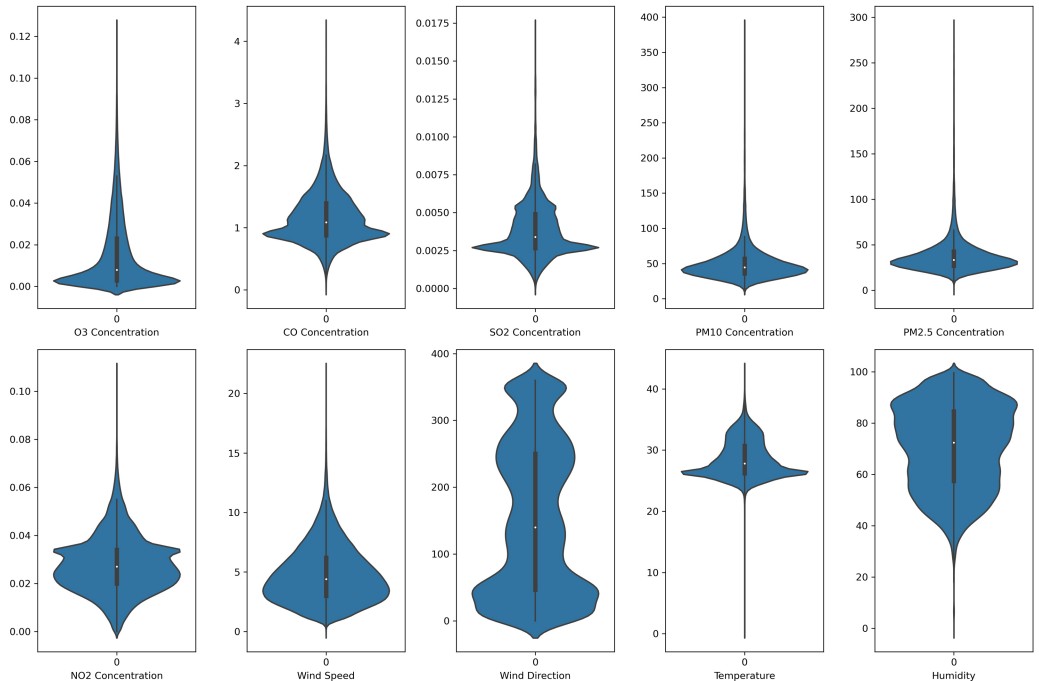

**Figure 3  Violin plot of the pollutants and meteorological data distribution in Petaling air monitoring station.**

Figure 6 shows the correlation matrix between pollutant markers and associated meteorological factors. The coefficient value varies from −1 to +1. Coefficient value descriptions are described in Table 3. For example, the correlation matrix shows that parameter such as $PM_{10}$ has a strong correlation with $PM_{2.5}$ while there are no parameters that show a strong negative correlation with $PM_{2.5}$. On the other hand, a parameter such as $O_3$, $SO_2$, $NO_2$, wind speed, and temperature are having a weak positive correlation with $PM_{2.5}$ while humidity and wind direction shows a weak negative correlation with $PM_{2.5}$.

$$r = \frac{\sum(x_i - \bar{x})(y_i - \bar{y})}{\sqrt{\sum(x_i - \bar{x})^2 \sum(y_i - \bar{y})^2}}$$

r = correlation coefficient
$x_i$ = x-variable parameters values in the dataset
$\bar{x}$ = mean of the values of the parameters
$y_i$ = y-variable parameters values in the dataset
$\bar{y}$ = mean of values of the parameters

## Population data

Selangor is the state with the largest economy in Malaysia resulting in a high population in the state with approximately 7,000,000. Each district in Selangor has its municipal council in taking care of and controlling the development of the district. From Fig. 1, the largest district in Selangor is Hulu Selangor while the smallest district is Petaling. However, in terms of population, Petaling has the highest population despite being the

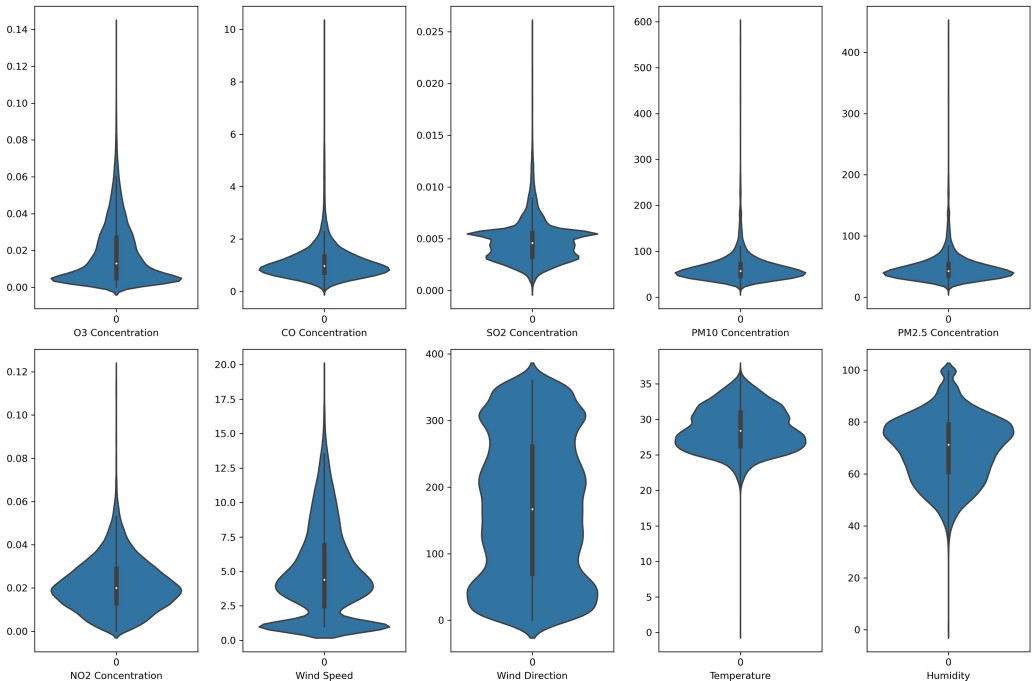

**Figure 4** **Violin plot of the pollutants and meteorological data distribution in Klang air monitoring station.**

smallest district area. In this study, the air monitoring station in Selangor is located in Petaling, Klang and Kuala Langat districts as these districts have higher populations due to the rapid urbanization and industrial activities.

## Method

In this study, we integrate the prediction model of air pollution with the optimized features of the long short-term memory (LSTM) algorithm to offer an AI-assisted framework. By simulating only significant features or parameters, we have enhanced the existing LSTM algorithm for high prediction accuracy with reduced training and testing time. We have tested supervised machine learning and deep learning models as shown in the overall proposed methodology framework in Fig. 7. In this research the data are pre-processed before being fed into the predictive model algorithm. The data is then randomly splitted into 2 sets which are the testing and training sets which consist of 30% and 70% of the data respectively. In this study, the machine learning and deep learning algorithm that were tested include (SVR), multi-layer perceptron (MLP) regressor, random forest, K-nearest neighbor (KNN), adaptive boosting (ADA Boost) and long short-term memory (LSTM). The performances of these models were evaluated using the performance matrix such as root mean square error (RMSE) and coefficient of determination ($R^2$). We also performed feature optimization by performing hyperparameter tuning to get the best fit model. The best fit model will further undergo
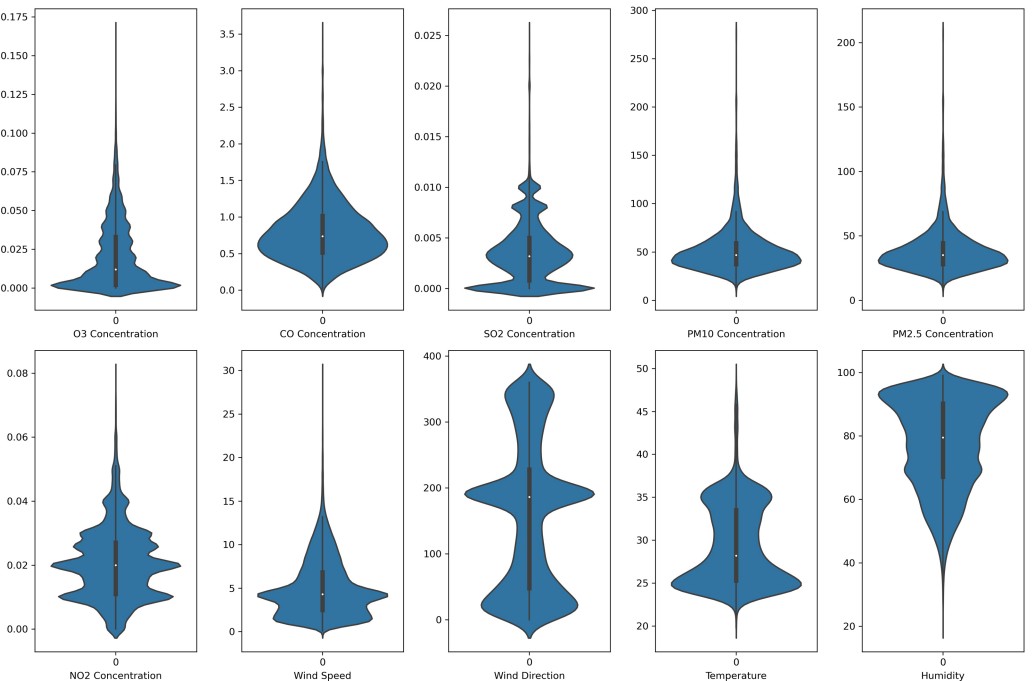

**Figure 5** Violin plot of the pollutants and meteorological data distribution in Shah Alam air monitoring station.

sensitivity analysis to identify the most significant parameter in the predictions before model deployment.

## Data pre-processing

Data collected from the Department of Environment (DoE) were filtered and preprocessed to remove irrelevant data. The dataset collected contains missing values which cause significant impacts to the accuracy of predictions. Therefore, it is crucial to perform data imputation in handling the missing values. In this study, interpolation was performed when filling in the missing values as the dataset has a continuous data characteristic which is suitable for interpolation imputation. After the missing value was filled, the datasets were further processed through normalized for the input and output parameters. Due to the wide range of data, normalization plays an important role in standardizing the dataset. The following Min-Max normalization approach equation was used in the normalization of the data.

$$x_{scaled} = \frac{x - min(x)}{max(x) - min(x)}$$

$x_{scaled}$ = normalized value
$x$ = observed value for normalization
$max(x)$ = dataset maximum value
$min(x)$ = dataset minimum value

**Table 2  Statistical analysis of the parameters in the 4 monitoring stations in Selangor.**

| | Banting | | | | | Petaling | | | | |
|---|---|---|---|---|---|---|---|---|---|---|
| | Min | Max | Mean | 50% | SD[a] | Min | Max | Mean | 50% | SD |
| PM$_{2.5}$ | 10.563 | 328.958 | 42.110 | 36.831 | 23.027 | 0.000 | 292.327 | 37.938 | 33.393 | 20.910 |
| PM$_{10}$ | 14.084 | 438.610 | 56.147 | 49.108 | 30.703 | 12.08 | 389.770 | 50.585 | 44.522 | 27.879 |
| NO$_2$ | 0.000 | 0.066 | 0.012 | 0.011 | 0.007 | 0.000 | 0.109 | 0.028 | 0.027 | 0.012 |
| SO$_2$ | 0.000 | 0.023 | 0.004 | 0.004 | 0.002 | 0.000 | 0.017 | 0.004 | 0.003 | 0.002 |
| CO | 0.000 | 5.658 | 0.638 | 0.579 | 0.329 | 0.016 | 4.250 | 1.162 | 1.089 | 0.411 |
| O$_3$ | 0.000 | 0.149 | 0.021 | 0.013 | 0.021 | 0.000 | 0.124 | 0.015 | 0.008 | 0.017 |
| Temperature | 0.000 | 37.200 | 26.526 | 25.900 | 3.772 | 0.000 | 43.500 | 28.557 | 27.800 | 3.039 |
| Humidity | 0.000 | 99.700 | 78.130 | 81.500 | 18.839 | 0.000 | 99.600 | 70.713 | 72.400 | 16.363 |
| Wind Speed | 0.000 | 89.100 | 4.564 | 4.000 | 4.322 | 0.000 | 22.000 | 4.811 | 4.400 | 2.389 |
| Wind Direction | 0.000 | 360.00 | 160.647 | 151.000 | 94.821 | 0.000 | 360.000 | 153.871 | 140.000 | 112.369 |

| | Klang | | | | | Shah Alam | | | | |
|---|---|---|---|---|---|---|---|---|---|---|
| | Min | Max | Mean | 50% | SD | Min | Max | Mean | 50% | SD |
| PM$_{2.5}$ | 10.277 | 446.333 | 48.403 | 42.914 | 26.377 | 7.440 | 211.230 | 38.731 | 35.070 | 17.463 |
| PM$_{10}$ | 13.703 | 595.110 | 64.537 | 57.219 | 35.169 | 9.920 | 281.640 | 51.642 | 46.760 | 23.284 |
| NO$_2$ | 0.000 | 0.121 | 0.022 | 0.020 | 0.012 | 0.000 | 0.080 | 0.021 | 0.020 | 0.011 |
| SO$_2$ | 0.000 | 0.026 | 0.004 | 0.005 | 0.002 | 0.000 | 0.026 | 0.004 | 0.003 | 0.003 |
| CO | 0.033 | 10.195 | 1.117 | 0.975 | 0.681 | 0.015 | 3.558 | 0.797 | 0.735 | 0.396 |
| O$_3$ | 0.000 | 0.141 | 0.018 | 0.013 | 0.017 | 0.000 | 0.166 | 0.020 | 0.012 | 0.021 |
| Temperature | 0.000 | 37.200 | 28.653 | 28.400 | 3.054 | 19.800 | 49.300 | 29.43 | 28.200 | 4.775 |
| Humidity | 0.000 | 99.600 | 70.033 | 71.300 | 12.607 | 19.900 | 99.000 | 77.43 | 79.533 | 14.330 |
| Wind Speed | 1.000 | 19.300 | 4.972 | 4.400 | 3.322 | 0.100 | 29.900 | 5.013 | 4.293 | 3.290 |
| Wind Direction | 0.000 | 360.000 | 169.092 | 167.000 | 106.735 | 0.000 | 360.000 | 158.770 | 186.310 | 110.017 |

**Notes.**
[a] SD, Standard Deviation.

**Table 3  Correlation coefficient range and descriptions.**

| Correlation Coefficient Range | Description |
|---|---|
| $0.75 \leq r \leq 1$ | Strong positive correlation |
| $-0.75 \leq r \leq -1$ | Strong negative correlation |
| $r \leq 0.25$ | Weak positive correlation |
| $r \leq -0.25$ | Weak negative correlation |

After the parameters are normalized, the dataset is splitted into two sets which are training and testing sets which consist of 70% and 30% of the datasets, respectively. The dataset splitting for training and testing are randomly assigned by the algorithm to avoid data bias. In addition, the dataset is labelled for input and output parameters for the training and testing procedure before feeding into the models. The input parameters

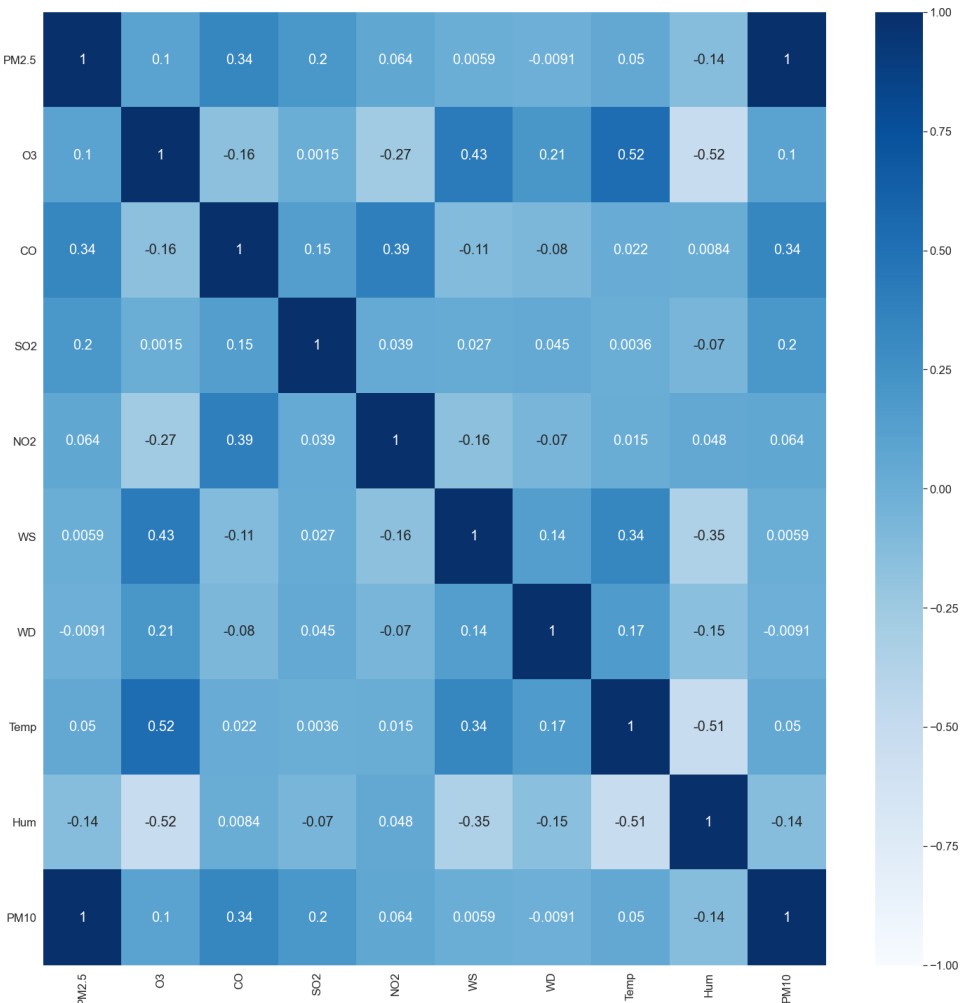

**Figure 6** Correlation matrix of the pollutant markers.

include all the air pollutant markers and meteorological parameters while the output parameter refers to the targeted air pollutant markers.

## AI algorithm modelling

In this article, we constructed supervised machine learning and deep neural network in performing air quality monitoring and prediction through regression models, using Python programming language. The regression algorithm allows the predictions of one or more predictors to output continuous outcomes, hence regression is suitable and able to predict the continuous output of the air quality. The regression models employed in this study include the multilayer perceptron (MLP) regression model, random forest regressor, adaptive boosting, support vector regressor (SVR), and k-nearest neighbor (KNN) regressor and the deep neural network involved is long short-term

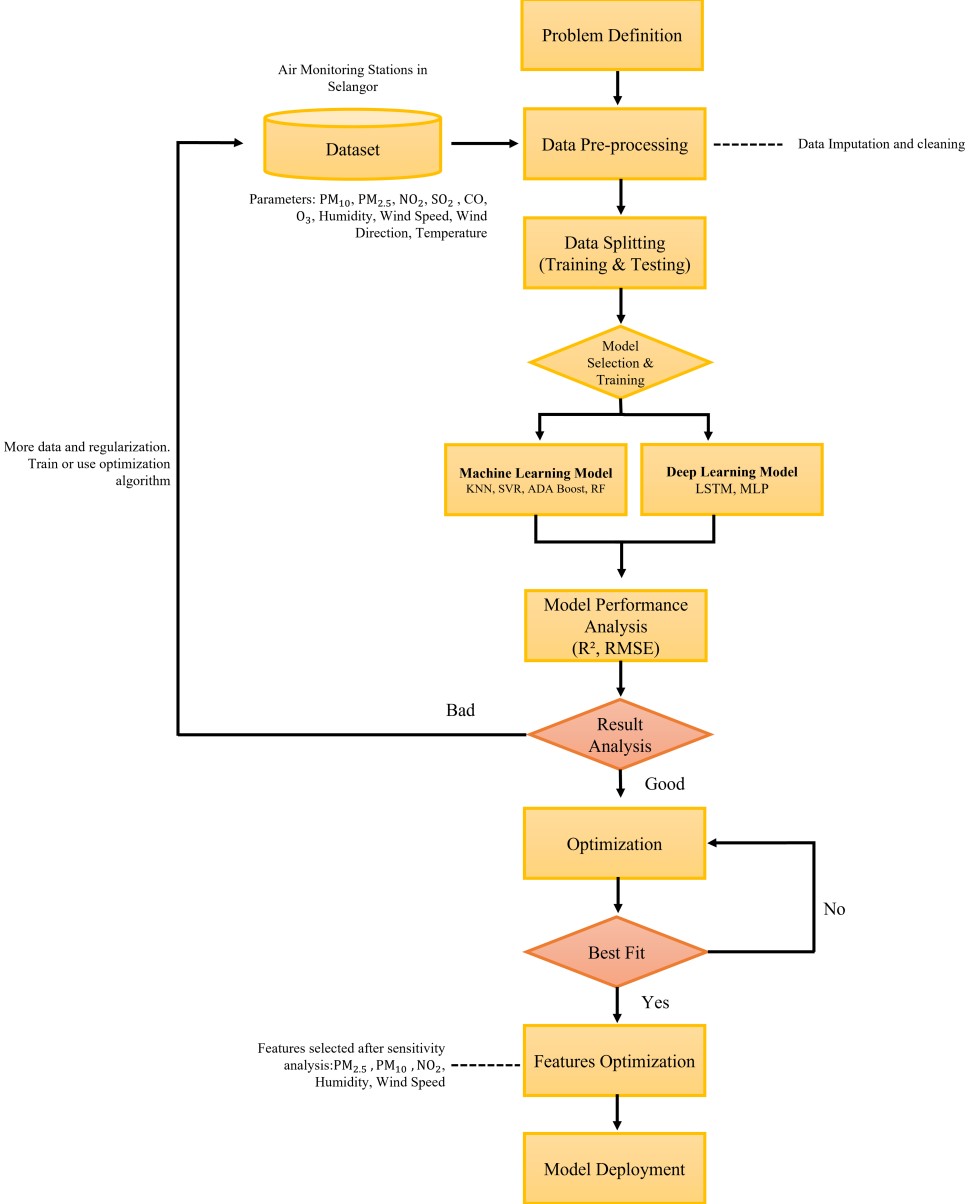

**Figure 7**  **The proposed methodology framework for the prediction of air pollutants.**

memory (LSTM). Table 4 describes the models involved in this study and LSTM model is described in the following sections.

## Model performance evaluation

To evaluate the performance and the fitness of the algorithm, statistical measurements namely i. root mean squared error (RSME), and ii. the coefficient of determination ($R^2$) is applied in analyzing the performance of the models. The mathematical expression of the

**Table 4   AI Models descriptions in the study.**

| Techniques | Description |
| --- | --- |
| **Adaptive boosting (AdaBoost)** | AdaBoost is typically an ensemble model, consisting of many base learners which typically outperform a single learner. By creating numerous regressors, AdaBoost regression may automatically alter the weighting of the model based on estimating mistakes. It is also has the capability in improving the generalization of nonlinear and complex regression problems (*Wong et al., 2022a*; *Zhang & Yang, 2018*). Weighting: $\lambda_t = \frac{1}{2}\ln\left(\frac{1-e_t}{e_t}\right)$ Model: $f_T(x) = \text{sgn}\left(\sum_{t=0}^{T-1}\lambda_t \cdot g_t\right)$ |
| **Support vector regressor (SVR)** | As generalization of support vector machines, the support vector regression can estimate the continuous output value (*Cervantes et al., 2020*; *Wong et al., 2022a*). The generalization approach borrowed from the structural risk minimization theory (SRM), reduces the empirical hazards of overfitting in statistical learning theory. Due to the advantages, SVR is well recognized in pattern recognition and regression application. Similar to SVM, SVR adopted the concept of hyperplane a separation boundary that will assist in the predictions and boundary line, a line other than hyperplane that creates margin, which act as a predictor tools. Hyperplane: $wx + b = 0$ Boundary line: $wx + b = \pm e$ |
| **K-nearest neighbor (KNN)** | K-nearest neighbor (KNN), a simple and well-known supervised machine algorithm is adopted in this study. KNN gathers data points by distances or radius from the arrival data point, where the radius can be measured in a variety of ways. The most recommended way of measuring the radius is the Euclidean distance, which as shown as the formula. $d(x,y)$ $= \|x - y\|$ $= \sqrt{(x-y)\cdot(x-y)}$ $= \left(\sum_{i=1}^{m}\left((x_i - y_i)^2\right)\right)^{1/2}$ |
| **Random forest** | Random forest is supervised machine learning where the decision trees are the foundation for the modelling predictions and analysis. Random forest is made up of numerous decision trees. Each prediction is made by averaging the result from different trees. The result of the predictions improved as the tree numbers increased. This technique brings several advantages such as less computation time, ease of working with high-dimensional data, and strong fault tolerance. The advantages of making it a superior algorithm to the other machine learning model. |

**Table 4** (*continued*)

| Techniques | Description |
| --- | --- |
| **Multilayer perceptron (MLP)** | MLP is a popular classification and regression variation of the standard ANN model (*Ay & Kisi, 2012*). It consists of a network of sigmoid activation neurons connected by links with various weights, which serve as the basis for the MLP model. The input layer of MLP passes via hidden levels to reach the output layer, making up the MLP model's three fundamental layers *Ay & Kisi (2012)*; *Wong et al. (2022a)*. The activation function receives the input from the preceding later and passes the output to the following later. $Net_i = b_i + \sum_{j=1}^{n} w_{ij} x_j$ $b_i$ = threshold x=input value $w_i$ and $w_{ij}$ = assigned weight that represents each neuron's strength Activation function: $f(Net_i) = \frac{1}{1+e^{-Net_i}}$ |
| **Long short-term memory networks (LSTM)** | LSTM is a state-of-the-art technique evolved from the recurrent neural network (RNN), proposed by Hochreiter and Schmidhuber, where it evolved from RNN. LSTM replicates the memory cell concept, enable the weights changes at the following instant without producing a disappearing or bursting gradient problem by using the output of the previous moment as the input of the subsequent moment. LSTM like-wise consists of 3 layers, which are one input layer, one output layer and a series of the hidden layer. What makes LSTM distinct from other neural networks is that it has hidden layers that are composed of one or more self-recurrent memory blocks where the blocks enable the preservation and subsequent retrieval of a value (forwards pass) or gradient (backward pass) that flows into the block at the necessary time step. |

calculation of the statistical measurement is expressed as follows:

$$RMSE = \sqrt{\frac{1}{n} \sum_{i=1}^{n} (y_i - \hat{y})^2}$$

$$R^2 = 1 - \frac{\sum_i (y_i - \hat{y})^2}{\sum_i (y_i - \bar{y})^2}$$

RMSE illustrates the square root of the average squared difference between the predicted and actual values, in other words, the square root of the estimated error. $R^2$ is the squared correlation between the predicted and actual datasets in regression models. The proportion variation of the results inferred by predictor factors is measured (*Wong et al., 2022a*). It determines the strength of the relationship between the model and the dependent variable. The higher the $R^2$, the fitter the model.

## Feature optimization

The purpose of this phase is to assess and rank the most important features of the air quality prediction model. Each predictive model's performance (developed in the previous stage) was compared, and the model with the best results was utilized to derive

**Table 5  Details information of LSTM model and parameters.**

| | Parameter | Description | Value | | | |
|---|---|---|---|---|---|---|
| | | | Banting | Petaling | Shah Alam | Klang |
| | N | Number of samples | 52,287 | 52,268 | 31,910 | 33,450 |
| Input | $\iota$ | Length of time | 24 | 24 | 24 | 24 |
| | c | Feature dimension | 10 | 10 | 10 | 10 |
| Output | N | Number of samples | 52,287 | 52,268 | 31,910 | 33,450 |

**Table 6  Experimental optimization setting of LSTM model.**

| Parameter | Values |
|---|---|
| Training set | 70% (randomly assigned by the algorithm) |
| Testing set | 30% (randomly assigned by the algorithm) |
| Maximum epochs of training | 20 |
| Patience of early stopping mechanism | 5 |
| Batch size | 32 |
| Window length | 24 |
| Learning rate | 0.001 |
| Loss function | Mean Squared Error (MSE) |

the feature optimization. The feature with the highest significant score (*i.e.,* R score) is the most significant predictor of the model. This process is performed on the best performance model algorithm which is in this study is LSTM. The final model of LSTM is further optimized using hyperparameter tuning as tabulated in Tables 5 and 6. The feature optimization process using sensitivity analysis framework is illustrated in Fig. 8. The predictors importance assessment which the predictor variables are ranks based on the prediction results. The analysis involved the input variable pruning (*Gazzaz et al., 2012*). The feature optimization was performed by testing the performance of the predictive model when input feature is removed one at a time. The indicator of the assessment is based on ranking of the ratio of errors upon the removal of the variables. The higher the error ratio indicates the higher significance of the parameter. In this study, the RMSE is being assessed as the error ratio. The higher the RMSE value, the more significant the parameter is. In this study, the features with RMSE values greater than 0.008 and R score greater than 0.55 is chosen and labelled as significant features in $PM_{2.5}$ prediction. Meanwhile, features with R score lesser than 0.55 indicated as less important and ignored in the final model prediction.

## RESULTS

In this section, the results of the predictive models are presented based on the monitoring site. We compared the performance of the AdaBoost, RF, SVR, KNN, LSTM and MLP predictive models. These models are trained to predict the pollutant markers in each station, based on the 10 input parameters as mentioned in the Methodology section. The

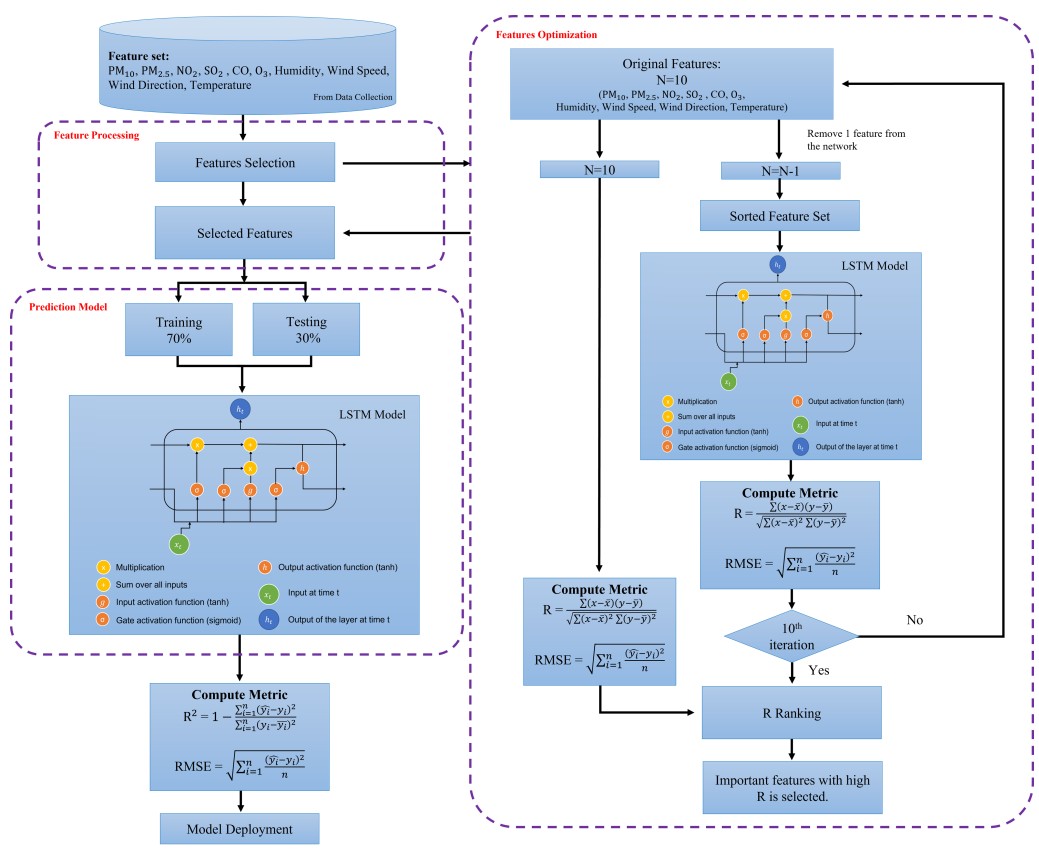

**Figure 8** **Feature optimization using sensitivity analysis framework.**

targeted pollutant markers are $PM_{10}$, $PM_{2.5}$, ozone ($O_3$) and carbon monoxide (CO). Table 7 presents the performances and predictive results of the models using machine learning and deep learning in Banting station, Petaling station, Shah Alam station and Klang station. Results and performances of the target pollution markers concentration prediction from each station are presented in this section and will be further discussed in the next section.

According to Table 7, the 4 pollution markers predicted for Banting station show an overall good result. In the CO concentrations predictions in Banting, LSTM attained the best prediction results with $R^2$ value of 0.725, followed by RF with an $R^2$ value of 0.663. Both techniques presented low RMSE, where LSTM has an RMSE 0.057 while RMSE in the RF is higher, 0.067. Hence, we can see that LSTM is a good predictor in CO concentration predictions. For $O_3$ concentration prediction, LSTM performed the best, followed by random forest and MLP. In the prediction, LSTM has an $R^2$ value of 0.889 and an RMSE of 0.053. while RF and MLP are having $R^2$ values of 0.771 and 0.708 respectively. RMSE of RF and MLP are 0.137 and 0.154. Meanwhile, in $PM_{10}$ prediction, all techniques attained good prediction results with $R^2$ values above 0.9. On the other hand in the prediction of $PM_{2.5}$, RF, LSTM and MLP are having a good performance, with an $R^2$ value greater than 0.99. LSTM has an $R^2$ value of 0.998, while MLP and

**Table 7 Performances of predictive models using machine learning and deep learning in air monitoring stations in Selangor.**

| | | Banting Station | | | | Petaling Station | | | | Klang Station | | | | Shah Alam Station | | | |
|---|---|---|---|---|---|---|---|---|---|---|---|---|---|---|---|---|---|
| | | CO | $O_3$ | $PM_{10}$ | $PM_{2.5}$ | CO | $O_3$ | $PM_{10}$ | $PM_{2.5}$ | CO | $O_3$ | $PM_{10}$ | $PM_{2.5}$ | CO | $O_3$ | $PM_{10}$ | $PM_{2.5}$ |
| ADA Boosting | $R^2$ | 0.463 | 0.460 | 0.978 | 0.978 | 0.095 | −0.275 | 0.974 | 0.983 | 0.400 | 0.269 | 0.971 | 0.971 | 0.353 | 0.284 | 0.985 | 0.989 |
| | RMSE | 0.086 | 0.21 | 0.022 | 0.022 | 0.184 | 0.317 | 0.024 | 0.019 | 0.103 | 0.202 | 0.020 | 0.020 | 0.262 | 0.311 | 0.022 | 0.018 |
| Random Forest | $R^2$ | 0.663 | 0.771 | 0.999 | 0.999 | 0.449 | 0.742 | 0.999 | 0.999 | 0.668 | 0.711 | 0.999 | 0.999 | 0.514 | 0.751 | 0.999 | 0.999 |
| | RMSE | 0.067 | 0.137 | 0.007 | 0.006 | 0.143 | 0.143 | 0.001 | 0.001 | 0.077 | 0.127 | 0.034 | 0.001 | 0.157 | 0.137 | 0.009 | 0.009 |
| MLP | $R^2$ | 0.527 | 0.708 | 0.999 | 0.999 | 0.280 | 0.688 | 0.998 | 0.998 | 0.488 | 0.652 | 0.996 | 0.996 | 0.351 | 0.535 | 0.998 | 0.998 |
| | RMSE | 0.081 | 0.154 | 0.006 | 0.006 | 0.164 | 0.156 | 0.006 | 0.006 | 0.095 | 0.139 | 0.007 | 0.007 | 0.181 | 0.187 | 0.008 | 0.008 |
| SVR | $R^2$ | 0.535 | 0.681 | 0.935 | 0.935 | 0.281 | 0.680 | 0.906 | 0.889 | 0.519 | 0.647 | 0.919 | 0.919 | 0.353 | 0.514 | 0.937 | 0.937 |
| | RMSE | 0.079 | 0.162 | 0.038 | 0.038 | 0.164 | 0.159 | 0.045 | 0.047 | 0.092 | 0.141 | 0.034 | 0.034 | 0.181 | 0.191 | 0.044 | 0.044 |
| KNN | $R^2$ | 0.478 | 0.696 | 0.946 | 0.946 | 0.313 | 0.665 | 0.924 | 0.923 | 0.572 | 0.643 | 0.912 | 0.909 | 0.432 | 0.623 | 0.931 | 0.921 |
| | RMSE | 0.084 | 0.158 | 0.035 | 0.035 | 0.16 | 0.162 | 0.041 | 0.039 | 0.087 | 0.141 | 0.035 | 0.036 | 0.170 | 0.169 | 0.046 | 0.047 |
| LSTM | $R^2$ | 0.725 | 0.889 | 0.997 | 0.998 | 0.937 | 0.713 | 0.856 | 0.995 | 0.894 | 0.818 | 0.947 | 0.918 | 0.542 | 0.589 | 0.813 | 0.993 |
| | RMSE | 0.057 | 0.053 | 0.014 | 0.010 | 0.024 | 0.061 | 0.006 | 0.010 | 0.014 | 0.036 | 0.016 | 0.058 | 0.039 | 0.049 | 0.010 | 0.012 |

random forest are having $R^2$ value of 0.999 and an RMSE value of 0.006. LSTM has an RMSE value of 0.010. Hence, we can conclude that the deep learning algorithm is suitable for the dataset in Banting. LSTM and MLP are having good performance although random forest has the highest performance value in particulate matters ($PM_{10}$ and $PM_{2.5}$) predictions. LSTM has a good result in CO and $O_3$ concentration which is the GHG predictions. Although random forest presented the best prediction in $PM_{10}$ and $PM_{2.5}$, it is tending to overfit. Therefore, the best algorithm for the prediction of pollutants in Banting is LSTM.

Meanwhile, the predictive analysis conducted for Petaling station, indicated that, for prediction of $PM_{2.5}$ and $PM_{10}$ concentration, all techniques have good results with low values of RMSE and high $R^2$ values. The best techniques for the predictions of $PM_{2.5}$ and$PM_{10}$ concentration is presented by RF with the highest $R^2$ value, 0.999 and the lowest RMSE score which are 0.001. In the prediction of CO concentration, LSTM outperformed with an $R^2$ value of 0.937 as compared to the other algorithm where the $R^2$ values are less than 0.500. The RMSE value of LSTM is the lowest among the algorithms which are 0.024. On the other hand, $O_3$ concentration prediction shows a different result where the RF and LSTM have a higher performance in the predictions where the $R^2$ values are 0.742 and 0.713 respectively. RMSE for the models is 0.143 and 0.061 respectively. Due to the lower RMSE value, LSTM is the best model for the prediction of $O_3$ concentrations.

In Klang station CO concentration prediction, LSTM has the best performance as shown in Table 7 with the highest $R^2$ value which is 0.894 and an RMSE value of 0.014. In $O_3$ concentration predictions, LSTM and random forest are having good performances as compared to other models. LSTM and random forest have an $R^2$ value of 0.818 and 0.711 respectively, while the RMSE values are 0.036 and 0.127 respectively. Hence. LSTM is selected as the best model in the prediction of $O_3$ concentration in Klang station. In the prediction of $PM_{10}$, MLP is selected as the best algorithm while in the prediction of $PM_{2.5}$, random forest is chosen as the best model. MLP presented the lowest RMSE value which is 0.007 as compared to LSTM which are 0.016 when predicting $PM_{10}$. On the other hand, the random forest has the lowest RMSE value which is 0.001 as compared to the other models. From the result shown, although random forest has a good result in PM concentration predictions, especially in $PM_{2.5}$, however, the almost perfect result shows the tendency of overfitting. Hence, deep learning models such as LSTM and MLP are more suitable for the predictions.

In Shah Alam, CO concentration prediction performed by LSTM is chosen as the best model in CO concentration prediction as it has the lowest RMSE and highest $R^2$ value which are 0.039 and 0.542 respectively. In $O_3$ concentration prediction, due to the lowest RMSE value which is 0.049, LSTM is selected as the best model in the prediction with higher $R^2$ value of 0.589. In $PM_{10}$ concentration prediction, the random forest has the best performance, with an $R^2$ value of 0.999 and RMSE value of 0.009. However, based on the results shown, the model tends to overfit, hence, LSTM is chosen as the best model of prediction. In $PM_{2.5}$ concentration prediction, MLP and LSTM are having good results where the $R_2$ values are 0.998 and 0.993, meanwhile, the values of RMSE of the models are 0.008 and 0.012. From the result shown, a deep learning algorithm, especially LSTM is
**Table 8  Feature optimization results and combination of input parameters in LSTM algorithm.**

| PM$_{2.5}$ | PM$_{10}$ | CO | SO$_2$ | NO$_2$ | O$_3$ | Wind Speed | Wind Direction | Humidity | Temperature | R | RMSE |
|---|---|---|---|---|---|---|---|---|---|---|---|
| * | | | | | | | | | | **0.985** | **0.012** |
| | * | | | | | | | | | **0.869** | **0.008** |
| | | | | | | | | * | | **0.638** | **0.009** |
| | | | | | | * | | | | **0.621** | **0.009** |
| | | | | * | | | | | | **0.555** | **0.008** |
| | | * | | | | | | | | 0.525 | 0.007 |
| | | | * | | | | | | | 0.512 | 0.008 |
| | | | | | | | * | | | 0.492 | 0.008 |
| | | | | | * | | | | | 0.425 | 0.008 |
| | | | | | | | | | * | 0.318 | 0.007 |
| | | | | | | | | | | 0.178 | 0.008 |

**Notes.**

[a]*, Represent the removed parameter in each dataset. Bold values represent the selected R and RMSE values in the prediction.

more suitable for the prediction of the pollutant datasets in Shah Alam station due to the lower RMSE values and higher R$^2$ values.

In an overview of the performance, as shown in Table 7, the random forest has the highest performance in PMs prediction with an R$^2$ value of 0.999 in all stations. However, this result shows the tendency of the algorithm to be overfitted where the R$^2$ value is approximately 1.00 and the RMSE value is very small, approximately 0. In the prediction of CO and O$_3$ concentrations, the model that performed the best is LSTM in all stations as it has the lowest RMSE and highest R$^2$ in the prediction as compared to the other models. Hence from the result findings, it is shown that in the predictions of air pollution markers, machine learning, especially ensemble models such as ADA Boosting and random forest and deep learning models for instance LSTM and MLP are good in the predictions. In this study, deep learning is more suitable for the predictions of the pollutants as it shows a better performance. Deep learning is more compatible with the datasets of this study as shown in the comparison above as deep learning models result in high R$^2$ values and low RMSE values where it does not tend to overfit, especially in the PM$_{2.5}$ prediction.

Figure 9 illustrated the scattered plot of PM$_{2.5}$ concentration prediction using LSTM. PM$_{2.5}$ concentration is plotted as it is the major parameter and indicator in air pollution monitoring. According to WHO and UNEPA, PM$_{2.5}$ contributes the greatest health threat to humans in air pollution, and it is often used as a metric in legal air quality systems. Hence, in this study, PM$_{2.5}$ concentrations are the major pollutant in the predictions, followed by greenhouse gases. From the figures shown, it is shown that the PM$_{2.5}$ concentration prediction performance by LSTM is high and precise. The predicted concentrations are highly accurate and the variation with observed concentrations is low.

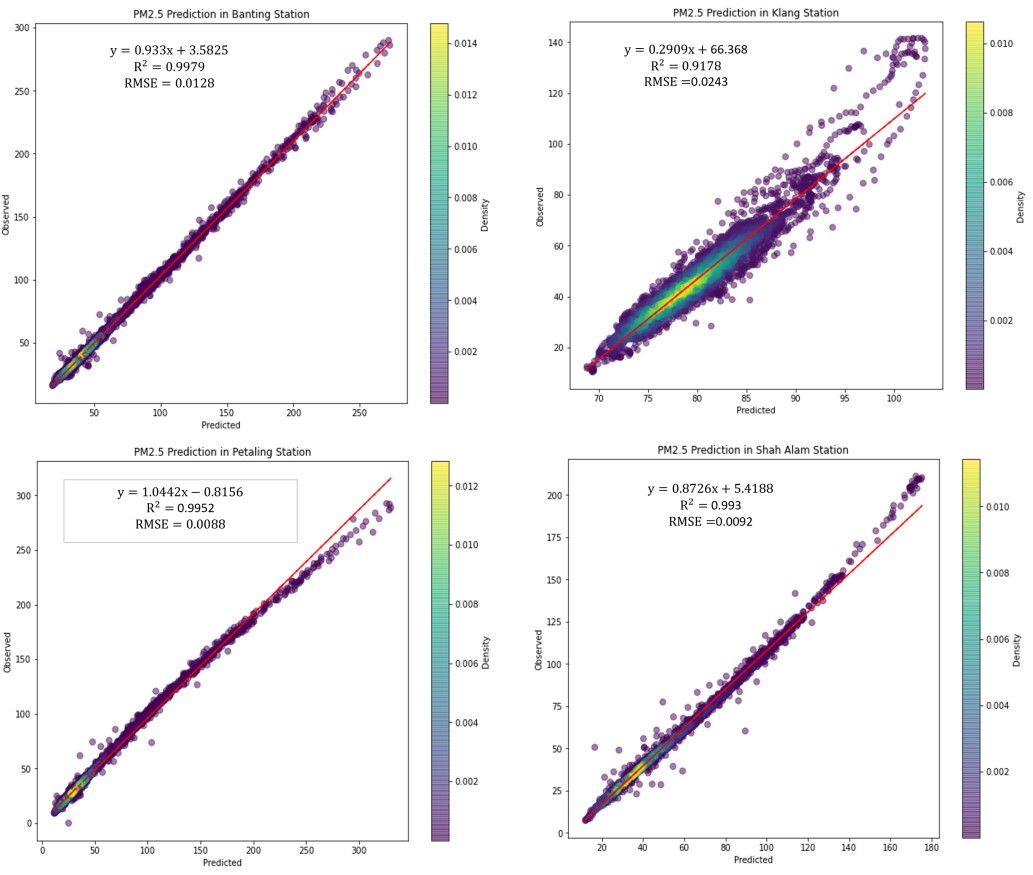

**Figure 9** Scattered plot of PM$_{2.5}$ concentration prediction using LSTM in the air monitoring stations in Selangor.

## Feature optimization

From the analyses presented above, LSTM has shown a great potential in achieving the highest R$^2$ and the lowest RMSE for major parts of the pollutant's prediction in almost all stations. To futher optimize the developed model, sensitivity analysis was performed using leave-one-out approach in the predictions to identify the most significant parameters that affect PM$_{2.5}$ LSTM prediction models. Table 8 tabulates the outcome of the sensitivity analysis where one of the parameters is excluded one after another from the modelling prediction. The influences of the input parameters and the PM$_{2.5}$ prediction performances are indicated by R and RMSE. From the results shown in Table 8, it indicates that PM$_{2.5}$ has the highest sensitivity ($R = 0.985$, RMSE $= 0.012$), followed by PM$_{10}$ ($R = 0.869$, RMSE $= 0.008$), humidity ($R = 0.638$, RMSE $= 0.009$), wind speed ($R = 0.621$, RMSE $= 0.009$), and NO$_2$ ($R = 0.555$, RMSE $= 0.008$). These are pollutants and meteorological factors are proven to highly affect the PM$_{2.5}$ concentration in the air. These parameters play an important role in PM$_{2.5}$ prediction using the LSTM model. After extracting the six significant parameters from the sensitivity analysis, we created a new LSTM model with only the six input parameters indicated above. The overall

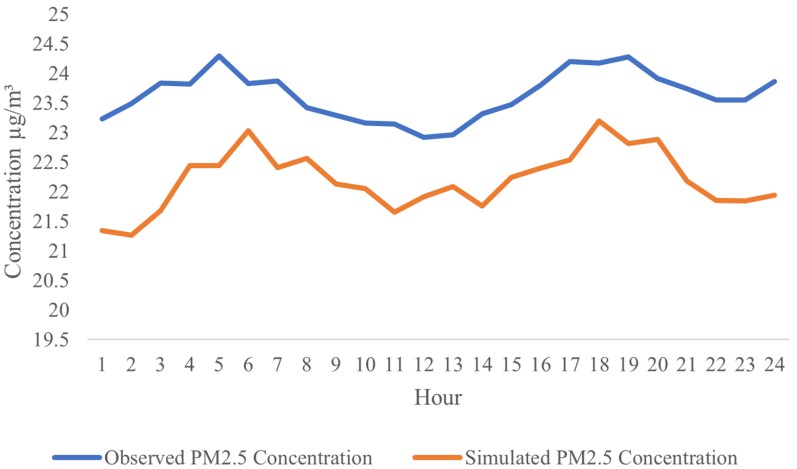

a.) PM$_{2.5}$ concentration prediction performance plot before features optimization.

.

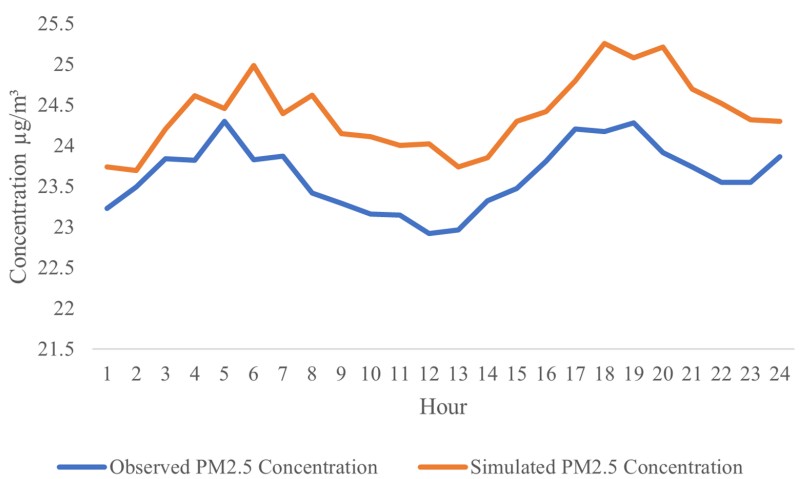

b.) PM$_{2.5}$ concentration prediction performance plot after features optimization.

.

**Figure 10** **Forecast a feature-optimized LSTM, original observation and predicted data.**

performance of the LSTM prediction model has enhanced the sensitivity analysis, with $R^2 = 0.9967$, RMSE $= 0.0045$ being replaced by $R^2 = 0.997$, RMSE $= 0.0046$ for PM$_{2.5}$ predictions with the five input parameters listed above. This demonstrates that, according to the sensitivity analysis, PM$_{10}$, PM$_{2.5}$, NO$_2$, wind speed and humidity are the primary determinants of PM$_{2.5}$ concentration. The five significant parameters prove to promote the high accuracy in PM$_{2.5}$ concentration prediction. We computed performance analysis on hourly data prediction of PM$_{2.5}$ using only the optimized features with an improved LSTM model as shown in Fig. 10. This figure depicts the good prediction performance of an improved LSTM model where the predicted PM$_{2.5}$ concentrations are in agreement with the observed PM$_{2.5}$ concentration.

## DISCUSSION

The preceding results demonstrate that the proposed feature-optimized LSTM model is the best model for prediction air pollution concentration. The prediction of pollutant concentrations is data-driven because it allows policy makers and government agencies to make strategic decisions based on the data analysis and interpretation of the pollutant concentration forecasts' input data. This is significant since it is one of the prerequisites for the emergence of smart cities and the fight against urban pollution. The government of Malaysia has started SmartSelangor, a smart city initiative in Selangor. As an additional effort to enhance the performance of the proposed predictive model, LSTM is tuned with features optimization, to identify the significant parameters. The tuning of the model is performed through sensitivity analysis, where the optimum features significantly affect the prediction performance are determined. As shown in Fig. 10, the predicted values after features optimization is relatively closer to the actual values as compared to the prediction results before optimization. In general case, prediction of air quality especially in air quality index (AQI) prediction required predictors such as PM, $SO_2$, $O_3$, CO, $NO_2$, and meteorological parameters. With optimized the tuned LSTM analysis and optimization in this model, the selected features are $PM_{10}$, $SO_2$, $NO_2$, temperature, humidity, and wind speed, from 10 predictors reduced to 5 predictors (*i.e.,* $PM_{2.5}$, $PM_{10}$, $NO_2$, humidity and wind speed). With the reduced number of predictors, the prediction of $PM_{2.5}$ could be performed without many interventions, especially in predictors data collections. In addition, it helps in reducing the cost and promotes low-cost air quality monitoring where the IoT sensors in pollutant concentration detection could be reduced. This can be supported by the lower training time (281 s) and testing time (41.18 s) as compared to model without optimization spent longer time in testing (55.22 s) and training (503 s). Furthermore, feature optimization has also improved the prediction accuracy of air quality models, as the model can focus on the most informative features, and avoid overfitting or noise. This can lead to more reliable and accurate air quality forecasts, which can be useful for public health, environmental monitoring, and city planning.

In addition, forecast and analysis of the population and $PM_{2.5}$ and $PM_{10}$ concentration for the four stations as shown in Fig. 11. All major cities indicated in this figure have similar population projections, where the population in these cities continued to grow and increase exponentially. In Shah Alam, the population was about 11,841 in 1900, and increased from 65,192 in 1960 to 100,262 in 1970. Meanwhile, in Klang, there is only 28,717 residences in 1900 which increased to 162,871 in 1960. A sudden surge in population happened in 1970 as the population raised to 252,283 from 162,871 within 10 years. Similar to Petaling, having a population of 50,546 in 1910, and increased to 211,758 in 1950. In 1960, the population of Petaling increased to 211,758. Since 2000, Shah Alam and Petaling were having a population change of 67.1% and 68% while 41.2% of the population change happened in Klang. Up to 2020, Petaling is having a population of 2,282,581, 632,638 in Shah Alam and 1,015,234 in Klang. The median age of the population in Petaling falls on the age of 27.2, age of 26.5 in Shah Alam and 27.1 years old in Klang. Looking at the median age, we can estimate that these cities are mainly

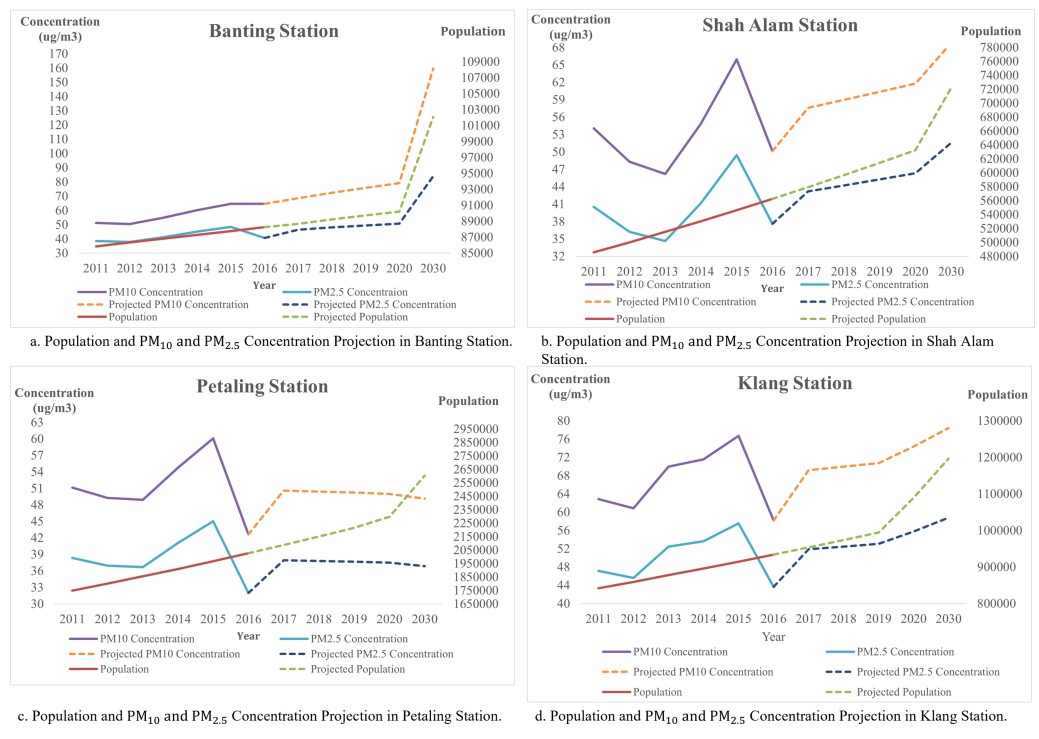

**Figure 11** (A–D) Population and $PM_{10}$ and $PM_{2.5}$ concentration projection of air monitoring station in Selangor.

occupied by young people. Hence, we could estimate the populations are young working adults, where transportation is highly needed. This can be supported by $CO_2$ emission which is approximately 19,358,692 tons per year in Petaling, and 5,361,626 tons per year in Shah Alam. In Klang, the $CO_2$ emitted in a year is approximately 6,514,216 tons per year. Selangor has contributed 54,360,379 tons of $CO_2$ per year to Malaysia.

In addition, The government of Malaysia intended to transform Selangor into a smart city, the Smart Selangor by 2025, producing a sustainable city at the same time to fit in the IR4.0 and digitalization era. With the tuned model presented in this article, it is possible to integrate the smart monitoring system with health impact assessment as proposed by the author to provide future strategies in data-driven smart city and smart healthcare system. This is crucial in maintaining sustainability during the urbanization and development of the economy, as a major contributor and element in building up the digitalized smart city and enable policymakers in rising solutions for development issues and maintenance of city sustainability. With the integrated system, a smart healthcare system could be established which could help in the early detection of potential impacts and potential groups of patients due to air pollution, aiding hospitals in preparing the resources. At the same time, the smart system can be embedded with edge devices where it could notify the air quality to the public as well as the government for mitigation steps. The proposed integrated plan is illustrated in Fig. 12. This research demonstrates that smart monitoring of air quality is applicable to the concept of smart cities. We have

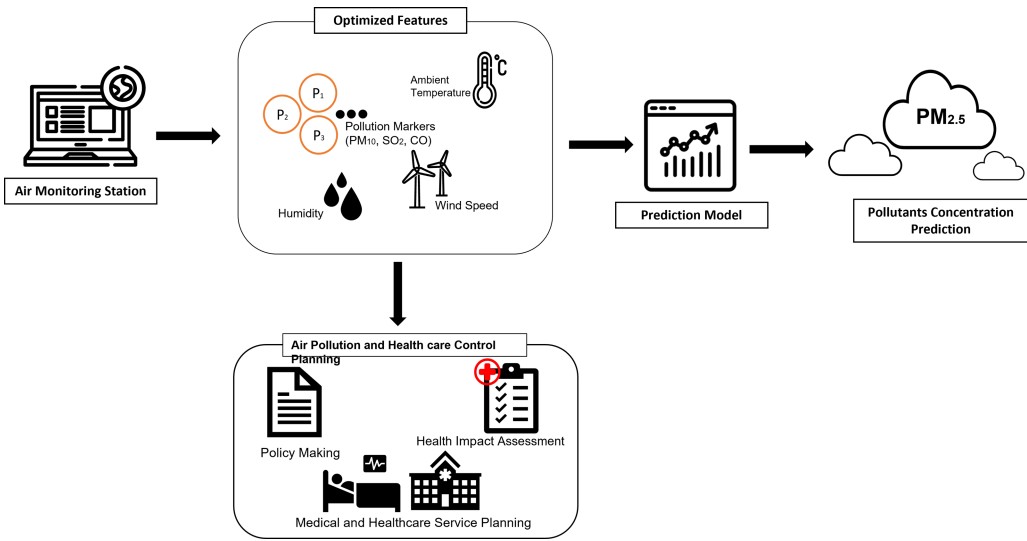

**Figure 12** **AI-assisted framework of integrated air pollution monitoring model.**

demonstrated that the ability of leverage big data in AI will not only aid policymakers in implementing smart monitoring but could also be extended to smart enforcement. Selangor is the largest economic state in the country, so it is understandable that the risk of air pollution is also higher due to a variety of economic and population-related activities. For a broader perspective of air quality, and to increase the coverage and build a more complex and comprehensive system, it is suggested to increase the research size to other states in Malaysia, to increase the pollution sources.

Many government policies aim to reduce greenhouse gas emissions, such as promoting the use of renewable energy sources, regulating emissions from industrial sources, and encouraging energy-efficient practices. These policies can lead to a reduction in greenhouse gas emissions, which can result in improved air quality. However, some policies aimed to reducing greenhouse gas emissions can also have unintended consequences for air quality. For example, policies that incentivize the use of biofuels may increase the production of crops, leading to increased emissions from agricultural machinery and fertilizers.Moreover, some governmental policies may focus primarily on reducing greenhouse gas emissions and may not take into account their impact on air quality. For instance, policies that promote the use of diesel engines in public transportation to reduce greenhouse gas emissions may lead to increased emissions of particulate matter, a harmful air pollutant. Therefore, it is essential to develop policies that consider both their impact on greenhouse gas emissions and air quality. This requires a comprehensive approach that considers various factors such as the source of emissions, the technology used, and the overall environmental impact. In this way, the government can promote a more sustainable and healthier environment for its citizens.Therefore, through this study, AI can be integrated with governmental policies regarding greenhouse gas emissions and air quality in several ways:

- AI can be used to analyze large amounts of data to identify patterns and trends in emissions and air quality. This can help policymakers make informed decisions and develop more effective policies to reduce emissions and improve air quality.
- In addition, AI can be used to optimize existing policies, for instance by identifying the most effective ways to allocate resources to reduce emissions or identifying the most efficient way to implement energy-efficient practices.
- AI can be used to develop new policies, for instance by using predictive analytics to anticipate the impact of future trends on greenhouse gas emissions and air quality, or by identifying areas where policies can be improved to achieve better outcomes.
- Finally, AI can be used to monitor and enforce existing policies, for instance by using sensors and real-time data analysis to track emissions from industrial sources or to monitor air quality in real-time as indicated in our proposed framework in Fig. 12.

## CONCLUSIONS

Building a clean and green city while developing the country demands careful planning to balance societal and economic needs, sustainability and feasible continuous outcomes. Urbanization and development caused life-threatening hazards such as environmental pollution and climate change and are not sustainable. Hence, aiming to combat environmental contamination issues while maintaining sustainability during urbanization, we propose an AI-assisted framework of air quality monitoring system that can be adopted in smart cities. In this research, we presented a predictive model for forecasting the air pollutant concentration in Selangor, the largest economic state with the higher population. With its size of economy and population, it represents a very good challenge for a prediction study on the quality of air, as well as determining the contents of pollutants at the test sites. From the predictive model, we found that feature-optimized LSTM performed the best in predicting air pollution by showing the highest $R^2$ values in predicting $PM_{10}$, $PM_{2.5}$, CO and $O_3$ concentration in Selangor. In addition, the optimized LSTM suggested $PM_{2.5}$, $PM_{10}$, $NO_2$, humidity and wind speed are the major predictors in $PM_{2.5}$ concentration prediction. Coincidently with the study direction, Smart Selangor is the state government initiative for implementation of smart city statewide, and therefore, we proposed remedies and detailing such as by implementing several technologies such as IoT, integrated health impact assessment, as well as drone technology to improve the accuracy of air quality prediction and enhance the smart system by promoting more benefits such as notifying publics on the air quality depending on the health condition of each individual. In conclusion, AI can play an essential role in integrating governmental policies regarding greenhouse gas emissions and air quality by providing valuable insights and tools to develop, optimize, and enforce policies that aim to promote a more sustainable and healthier environment.

However, this study can be further improved by including more states for more data generalization. Next, natural phenomenon and extreme weather are not considered in this study. Thirdly, the shortcoming of using a real-time data source that is inconsistent leads to incomplete data collection. In the future, we suggest embedding the air quality

predictive models with IoT and edge computing systems, to increase the accuracy in analyzing the data in real-time rapidly. In addition, digital twining also can be implemented especially in health impact assessments to monitor and predict patients' well-being. Environmental quality systems can be embedded and linked to healthcare services using IoT, to enable the public to monitor the environment quality and assess the impacts on health, hence, examining the needs for healthcare services.

## ACKNOWLEDGEMENTS

The authors would like to thank the Department of Environment (DoE) Malaysia for the data provided in this study.

### Funding

This research is funded by Ministry of Higher Education through MRUN Young Researchers Grant Scheme (MY-RGS), MR001-2019, entitled "Climate Change Mitigation: Artificial Intelligence-based Integrated Environmental System for Mangrove Forest Conservation". The funders had no role in study design, data collection and analysis, decision to publish, or preparation of the manuscript.

### Grant Disclosures

The following grant information was disclosed by the authors:
Ministry of Higher Education through MRUN Young Researchers Grant Scheme (MY-RGS): MR001-2019.
Climate Change Mitigation: Artificial Intelligence-based Integrated Environmental System for Mangrove Forest Conservation.

### Competing Interests

The authors declare there are no competing interests.

### Author Contributions

- En Xin Neo conceived and designed the experiments, performed the experiments, analyzed the data, performed the computation work, prepared figures and/or tables, authored or reviewed drafts of the article, and approved the final draft.
- Khairunnisa Hasikin conceived and designed the experiments, performed the experiments, analyzed the data, performed the computation work, prepared figures and/or tables, authored or reviewed drafts of the article, and approved the final draft.
- Khin Wee Lai conceived and designed the experiments, performed the experiments, analyzed the data, performed the computation work, prepared figures and/or tables, authored or reviewed drafts of the article, and approved the final draft.
- Mohd Istajib Mokhtar conceived and designed the experiments, performed the experiments, analyzed the data, performed the computation work, prepared figures and/or tables, authored or reviewed drafts of the article, and approved the final draft.

- Muhammad Mokhzaini Azizan conceived and designed the experiments, performed the experiments, analyzed the data, performed the computation work, prepared figures and/or tables, authored or reviewed drafts of the article, and approved the final draft.
- Hanee Farzana Hizaddin conceived and designed the experiments, performed the experiments, analyzed the data, performed the computation work, prepared figures and/or tables, authored or reviewed drafts of the article, and approved the final draft.
- Sarah Abdul Razak conceived and designed the experiments, performed the experiments, analyzed the data, performed the computation work, prepared figures and/or tables, authored or reviewed drafts of the article, and approved the final draft.
- Yanto conceived and designed the experiments, performed the experiments, analyzed the data, performed the computation work, prepared figures and/or tables, authored or reviewed drafts of the article, and approved the final draft.

## Data Availability

The raw data is available in the Supplementary File.

## Supplemental Information

Supplemental information for this article can be found online at http://dx.doi.org/10.7717/peerj-cs.1306#supplemental-information.

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
