# Peer review of "Artificial intelligence-assisted air quality monitoring for smart city management"

_PeerJ Computer Science, doi:10.7717/peerj-cs.1306_

## Round 0.1 · original submission · Major Revisions

Based on the reviewers’ comments, you may resubmit the revised manuscript for further consideration. Please consider the reviewers’ comments carefully and submit a list of responses to the comments along with the revised manuscript.

Reviewer 1 ·

Basic reporting

The paper presents a study for forecasting air pollutants based on the dataset gathered from the region of Selangor. The investigation has used 4 machine learning and 2 deep learning techniques for prediction and claims that the results will help to build a holistic strategy to tackle air quality index.

The revised version of the paper has addressed several reviews of the previous cycle however authors should look into a few more aspects.

The authors have attempted to provide the motivation of the work however it still does not answer, who will benefit from this work and how? how this work differentiates itself from existing air quality prediction studies? This discussion should be added in section 1.

Experimental design

Figure & suggest that problem definition is extracted from the dataset. If the problem is dependent on the dataset then one should be able to modify the datset to solve it. Additionally there is not section describing a crisp problem statement.

The performance of ML models is highly dependent on quality of Dataset. Details of data preprocessing are not provided. Fig 7 suggest 5 Ml and 1 deep learning model that contradicts the earlier claim.

Authors have not provided the training and testing parameters for any of the models used for the study.

Validity of the findings

The figures 2-5 are not illegible. Also the labels of the correlation figure are easy to read.

Authors have not described the reasons behind the projected population increasing exponentially after 2020 in Banting in Fig 11. Authors have discussed that this has happened in the past but there is no evidence provided. The other stations show erratic bumps in the readings for multiple years.

Other than population, industrial and economic growth =, governmental policies regarding green house mission also affect the air quality index. Authors should consider including these features as well.

Reviewer 2 ·

Basic reporting

no comment

Experimental design

1. The research question is not well defined. From my humble view and the code the authors provided, they are not trying to solve a time series forecasting problem. Instead, they are trying to predict the PM2.5 value from other variables collected in the same timepoint. The authors should explicitly state the input dimension, e.g., which I think is number of variables*1 timepint.
2. If the above statement is true, the usage of LSTM is flawed and meaningless. LSTM is designed for capturing the temporal dependency in the time series forecasting problem. If only one time step is used, a simple feed-forward deep neural network would be enough.
3. The authors fail to justify how their research helps to fill in the knowledge gap when the considered problem is meaningless. The problem only happens when the PM2.5 sensor is broken, while the other sensors work. The adopted machine learning models are all well-known techniques, too.

Validity of the findings

Meaningful replication is not possible when the code for LSTM is not provided and the LSTM result is not reliable.

---

## Round 0.2 · accepted · Accept

The revisions are satisfactory and the manuscript is recommended for publication.

Reviewer 2 ·

Basic reporting

no comment

Experimental design

no comment

Validity of the findings

no comment

Additional comments

Dear authors,
Thanks for revising the manuscript. No further questions.